# Post-Gating Bias: Restoring Affine Freedom in Transformer MLPs

## Abstract

Modern transformers often omit additive biases because normalization and attention preserve constant offsets that downstream linear maps can remap. The gated MLP (SwiGLU) is a notable exception: the elementwise product with a nonlinear gate destroys such offsets, removing affine freedom after the nonlinearity. We examine a simple modification—Post-Gating Bias (PGB), an additive term applied after the gated product and before the down-projection. PGB restores this degree of freedom with negligible computational cost. Our working hypothesis is that PGB mitigates training noise from dropout or from additive stochastic regularization (e.g., in VAEs) by shifting activation boundaries in a controlled way, thereby softening sharp transitions that otherwise amplify perturbations. We observe stability gains at higher learning rates and some robustness improvements in a ViT-VAE setting. We also show other settings where the effect is minimal. We present these observations to clarify where biases are redundant in transformer blocks and where multiplicative gating makes them potentially useful. Finally, we report a controlled study varying dropout and latent noise across multiple seeds to test this hypothesis.

## 1 Introduction

Transformer architectures rely heavily on normalization layers to stabilize training. Because layer normalization (LN) and RMS normalization (RMSNorm) both incorporate additive biases, it is often assumed that explicit bias terms in intermediate linear transformations are redundant: any constant offset can be reproduced elsewhere in the block at negligible cost to expressivity. This assumption has motivated widespread omission of biases in recent transformer designs.

We revisit this assumption in the context of gated multilayer perceptrons (MLPs), the feed-forward design popularized by Shazeer (2020) and now used in nearly all state-of-the-art large language models. SwiGLU, in particular, is the standard choice in PaLM (Chowdhery et al., 2022), LLaMA (Touvron et al., 2023), Phi-3 (Abdin et al., 2024), and Gemma-3 (Team, 2024). While the redundancy of biases generally holds throughout most of the transformer block, it breaks down at a critical point: the final down-projection after elementwise gating. At this stage, signal modulation by the gate prevents constant offsets from being trivially reconstructed, so omitting a bias can interact with dropout or latent noise in ways that destabilize training.

To address this, we introduce a *post-gating bias* (PGB), added after the gated multiplication but before the down-projection:

$$\boldsymbol{U} = \boldsymbol{X}\boldsymbol{W}_u, \tag{1}$$

$$\boldsymbol{G} = \boldsymbol{X}\boldsymbol{W}_g, \tag{2}$$

$$\boldsymbol{Y} = \mathrm{dropout}(\boldsymbol{U}) \odot \mathrm{silu}(\boldsymbol{G}) + \boldsymbol{b}, \tag{3}$$

$$\boldsymbol{Z} = \boldsymbol{Y}\boldsymbol{W}_d. \tag{4}$$

This modification incurs virtually no additional parameter or memory cost, as the bias can be applied in place, yet it can smooth optimization when stochastic noise is present.

**Contributions.** Our work makes three contributions: (i) we revisit the assumption that biases in transformer blocks are redundant, providing an analysis of when biases can be absorbed into weight

matrices and showing that this equivalence fails after multiplicative gating; (ii) we propose the *post-gating bias* (PGB), a negligible-cost addition that restores affine freedom at the down-projection stage; (iii) we present preliminary experiments in three settings—a ViT-VAE on CelebA (Lee et al., 2020; Chang et al., 2022), modified Phi-3 language models (Abdin et al., 2024), and gated MLP classifiers—which suggest that PGB can improve stability in noisy latent settings and may enhance adversarial robustness, while showing little to no effect in large-scale autoregressive pretraining.

These findings indicate that even apparently redundant architectural elements can play subtle roles under stochastic training conditions, motivating further exploration in settings where noise and dropout strongly shape optimization dynamics.

## 2 BACKGROUND AND RELATED WORK

Early work on gated architectures demonstrated the effectiveness of gating in recurrent models (Jozefowicz et al., 2015; 2016), inspiring Dauphin et al. (2017) to introduce gated linear units (GLUs) in convolutional networks. Their formulation $(\boldsymbol{XW} + \boldsymbol{b}) \odot \sigma(\boldsymbol{XV} + \boldsymbol{c})$ included biases on all affine maps, and gating occurred in the same dimensionality as the input, with residual connections across convolutional blocks.

Building on this idea, Shazeer (2020) introduced GLU variants for transformer feed-forward networks, including ReGLU, GEGLU, and SwiGLU. Their design uses an explicit up–gate–down projection structure, but notably omits all bias terms in the MLP layers, including the down-projection. This bias-free formulation has since influenced large-scale LLMs, where biases in dense layers are often removed entirely. Subsequent works such as So et al. (2021) integrated gating variants into architecture search, and Ramesh & Ramkumar (2023) extended gated activations to attention blocks in vision transformers. While these contributions reinforce the importance of gating in transformer design, none have revisited the role of bias placement relative to dropout and gating.

In parallel, several lines of work have explored transformer-based VAEs. Conditional generation models such as T-CVAE (Wang et al., 2019) and TRACE (Hu et al., 2022) adapt variational bottlenecks for diverse text generation. Other approaches embed latent variables directly in transformer attention layers via nonparametric variational information bottlenecks (Henderson & Fehr, 2022), or construct structured latent spaces such as graph-induced syntactic-semantic embeddings (Zhang et al., 2023). Beyond NLP, transformer–VAE hybrids have been proposed for manifold-aware vision representation learning (Shamsolmoali et al., 2023) and reduced-order modeling in fluid dynamics (Solera-Rico et al., 2023). Collectively, these works show the versatility of combining transformers with variational inference, but they do not address how architectural details like bias and dropout placement in gated MLP blocks affect optimization stability.

Our experimental study confirms the prevailing view in language modeling—biases after gating do not improve perplexity in standard transformer LMs—but we also find evidence in autoencoding settings that post-gating biases can stabilize training under latent noise. This suggests their utility may depend not on expressivity but on the interaction between dropout, gating, and the statistical structure of the task.

## 3 THEORETICAL ANALYSIS

The work above suggests that bias placement does not affect model expressivity in most parts of the transformer block, but that gating combined with dropout or latent noise creates an exception: offsets routed through the multiplicative interaction become unstable.

To make this precise, we analyze when biases can be reconstructed from weights alone and when such reconstruction becomes ill-conditioned. We show that *(i)* constant offsets can be re-encoded throughout attention and other linear maps, *(ii)* this equivalence breaks down after multiplicative gating, where the mean direction itself carries variance, and *(iii)* a post-gating bias restores conditioning by centering the regressors seen by the down-projection. Proofs are deferred to Appendix A.

### 3.1 OUTPUT-BIAS RECONSTRUCTABILITY

We compare two realizations of the same affine map,

$$\boldsymbol{y}_1 = \boldsymbol{W}(\boldsymbol{x} + \boldsymbol{b}_{\text{in}}) + \boldsymbol{b}_{\text{out}} \quad \text{vs.} \quad \boldsymbol{y}_2 = \hat{\boldsymbol{W}}(\boldsymbol{x} + \boldsymbol{b}_{\text{in}}), \qquad \hat{\boldsymbol{W}} = \boldsymbol{W} + \boldsymbol{\Delta},$$

and ask for the *smallest* perturbation $\boldsymbol{\Delta}$ that reconciles the missing intercept.

**Exact-fit with minimal change.** We enforce agreement on the baseline direction and minimize the change:

$$\min_{\boldsymbol{\Delta}} \ \|\boldsymbol{\Delta}\|_F \quad \text{s.t.} \quad \boldsymbol{\Delta}\, \boldsymbol{b}_{\text{in}} = \boldsymbol{b}_{\text{out}}. \tag{5}$$

**Lemma 1** (Minimal-change exact-fit is rank-1)**.** *The unique minimizer of equation 5 is*

$$\boldsymbol{\Delta}^* = \frac{\boldsymbol{b}_{\text{out}}\, \boldsymbol{b}_{\text{in}}^{\top}}{\|\boldsymbol{b}_{\text{in}}\|_2^2}, \qquad \hat{\boldsymbol{W}} = \boldsymbol{W} + \boldsymbol{\Delta}^*.$$

*This perturbation exactly recovers the output bias and leaves all components of $\boldsymbol{x}$ orthogonal to $\boldsymbol{b}_{\text{in}}$ unaffected. Residual discrepancies are confined to the $\boldsymbol{b}_{\text{in}}$ direction and $\|\boldsymbol{\Delta}\|_F$ is minimal.*

**Operator-norm remark.** Among all $\boldsymbol{\Delta}$ satisfying $\boldsymbol{\Delta}\boldsymbol{b}_{\text{in}} = \boldsymbol{b}_{\text{out}}$, the rank-1 map in Lemma 1 minimizes all spectral norms. Any action off $\text{span}(\boldsymbol{b}_{\text{in}})$ would increase the operator norm and the worst-case discrepancy.

The construction above shows that weight-only fixes *exist*. However, first-order training does not target equation 5 explicitly. In practice, SGD may not discover $\boldsymbol{\Delta}^*$, and when dropout or latent noise is present, routing the intercept through $(\boldsymbol{x} + \boldsymbol{b}_{\text{in}})$ couples it to high-variance, intermittently masked directions, degrading conditioning and stability. A post-gating bias supplies the intercept directly, avoiding this mean-through-weights pathway.

*Remark* (Minimum-MSE perturbation)*.* Let $\boldsymbol{x}$ have mean 0 and covariance $\boldsymbol{\Sigma} \succeq 0$. The solution of

$$\min_{\boldsymbol{\Delta}} \ \mathbb{E} \left\| (\boldsymbol{W}(\boldsymbol{x} + \boldsymbol{b}_{\text{in}}) + \boldsymbol{b}_{\text{out}}) - (\boldsymbol{W} + \boldsymbol{\Delta})(\boldsymbol{x} + \boldsymbol{b}_{\text{in}}) \right\|_2^2$$

is

$$\boldsymbol{\Delta}^* = \boldsymbol{b}_{\text{out}}\, \boldsymbol{b}_{\text{in}}^{\top} (\boldsymbol{\Sigma} + \boldsymbol{b}_{\text{in}}\boldsymbol{b}_{\text{in}}^{\top})^{-1}, \quad \mathbb{E}\|\boldsymbol{y}_1 - \boldsymbol{y}_2\|_2^2\big|_{\min} = \|\boldsymbol{b}_{\text{out}}\|_2^2 \Big(1 - \boldsymbol{b}_{\text{in}}^{\top}(\boldsymbol{\Sigma} + \boldsymbol{b}_{\text{in}}\boldsymbol{b}_{\text{in}}^{\top})^{-1}\boldsymbol{b}_{\text{in}}\Big).$$

As $\boldsymbol{\Sigma} \to 0$, this reduces continuously to the rank-1 exact-fit in Lemma 1. Despite existence, SGD has no reason to recover this projection; hence we defer the proof to Appendix A and emphasize the stability benefits of an explicit post-gating bias in the main text.

**Reconstructability through attention and the output projection.** Let $\hat{\boldsymbol{X}} = \text{RMSNorm}(\boldsymbol{X}) + b$ and $[\boldsymbol{Q}, \boldsymbol{K}, \boldsymbol{V}] = \hat{\boldsymbol{X}}[\boldsymbol{W}_q, \boldsymbol{W}_k, \boldsymbol{W}_v]$. Any constant offset $b$ can be linearly propagated into $\boldsymbol{Q}, \boldsymbol{K}, \boldsymbol{V}$. Crucially, if $\boldsymbol{V}$ carries a constant offset per token, i.e., $\boldsymbol{V} = \hat{\boldsymbol{V}} + \mathbf{1}\, \boldsymbol{c}^{\top}$ for some $\boldsymbol{c} \in \mathbb{R}^{d_v}$ (with $\mathbf{1}$ the all-ones column), then with standard attention

$$\boldsymbol{P} = \text{softmax}\Big(\frac{\boldsymbol{Q}\boldsymbol{K}^{\top}}{\sqrt{d_k}}\Big) \quad (\text{row-stochastic: } \boldsymbol{P}\mathbf{1} = \mathbf{1}),$$

the attention output is

$$\boldsymbol{A} = \boldsymbol{P}\boldsymbol{V} = \boldsymbol{P}\hat{\boldsymbol{V}} + \boldsymbol{P}(\mathbf{1}\,\boldsymbol{c}^{\top}) = \boldsymbol{P}\hat{\boldsymbol{V}} + \mathbf{1}\,\boldsymbol{c}^{\top}.$$

Thus the attention block *exactly* transmits one full bias vector per row via $V$ because the softmax weights are row-normalized. Consequently, the final output projection

$$\boldsymbol{Z} = \boldsymbol{A}\boldsymbol{W}_o = (\boldsymbol{P}\hat{\boldsymbol{V}})\boldsymbol{W}_o + \mathbf{1}\,(\boldsymbol{c}^{\top}\boldsymbol{W}_o)$$

can also reconstruct a constant offset if needed. This reinforces that biases are readily recoverable *throughout* attention; the non-reconstructability we highlight pertains specifically to the multiplicative gating stage prior to the down-projection in the MLP.

## 3.2 THE GATING EXCEPTION

The expressivity equivalence breaks down after multiplicative gating. In the MLP we write

$$U = XW_u, \qquad G = XW_g, \qquad S = U \odot \mathrm{SiLU}(G), \qquad Y = S \odot M + b, \qquad Z = YW_d,$$

with an optional dropout mask $M$ and optional post-gating bias $b$. If $b = 0$, the down-projection $W_d$ is the only route to produce offsets. Unlike attention, the multiplicative interaction $U \odot \mathrm{SiLU}(G)$ destroys constant offsets; routing an offset through $S$ couples it to high-variance, intermittently masked directions.

**Ill-conditioning of weight-only reconstruction.** The reasoning of Lemma 1 and the minimum-MSE remark applies not only to strict constant offsets but also to reconstruction of the *expected* output. In this case, the minimal-change solution is again rank-1, aligned with the mean of the post-gating features. However, variation along the mean direction is typically larger than in other coordinates, since when $\mathrm{SiLU}(G)$ vanishes for a fraction of cases, the surviving coordinates both carry the expectation and accumulate higher variance. In high-dimensional latent spaces, information can ordinarily be encoded orthogonally to a bias direction, but under multiplicative gating the nonzero expectation itself *is* the signal, leaving no orthogonal subspace to carry offsets. Thus the rank-1 correction introduces a top-of-spectrum mode precisely in a high-variance direction, inflating the condition number of $W_d$ and making outputs acutely sensitive to small perturbations. This explains why weight-only recovery of intercepts is particularly ill-conditioned in gated MLPs.

**Proposition 1** (Intercept removal improves conditioning). *Let $\varphi \in \mathbb{R}^p$ denote the vectorized post-gating features (e.g., a row of $S \odot M$), with mean $\mu = \mathbb{E}[\varphi]$ and covariance $\Sigma = \mathrm{Cov}(\varphi)$. Under squared loss, the (block) Hessian with respect to $W_d$ without an explicit bias term is*

$$H_{\mathrm{no\text{-}bias}} = \mathbb{E}[\varphi\varphi^\top] = \Sigma + \mu\mu^\top,$$

*while with a trainable post-gating bias $b$ that absorbs the mean offset, the effective Hessian governing $W_d$ is*

$$H_{\mathrm{pgb}} = \mathrm{Cov}(\varphi) = \Sigma.$$

*Since $H_{\mathrm{no\text{-}bias}} = \Sigma + \mu\mu^\top$ adds a rank-1 positive semidefinite term, its largest eigenvalue satisfies*

$$\lambda_{\max}(H_{\mathrm{no\text{-}bias}}) \geq \lambda_{\max}(\Sigma) + (\mathbf{v}^\top \mu)^2$$

*where $\mathbf{v}$ is the normalized top-eigenvector of $\Sigma$; consequently*

$$\kappa(H_{\mathrm{no\text{-}bias}}) \ \geq \ \kappa(\Sigma) \quad \text{with strict inequality when } \mu \notin \ker(\Sigma),$$

*so centering via an explicit intercept strictly improves (or leaves unchanged) the condition number.*

*Implication.* Without PGB, $W_d$ must learn using the uncentered design matrix $\mathbb{E}[\varphi\varphi^\top] = \Sigma + \mu\mu^\top$, whose additional rank-1 term both enlarges the top eigenvalue and—under dropout/latent noise—*fluctuates across steps*, creating a moving ill-conditioned direction. With PGB, the intercept is parameterized directly; gradients for $W_d$ are driven by centered features with Hessian $\Sigma$, yielding better conditioning and lower gradient variance.

**Variance control via post-gating bias.** Let $T$ denote the (per-token) training target in the decoder's space. Under a linear readout $Z = YW_d = (S \odot M)W_d + bW_d$, the optimal intercept for MSE satisfies

$$b^* W_d = \mathbb{E}[T] - \mathbb{E}[S \odot M] \, W_d.$$

At this solution, the residual driving updates of $W_d$ are

$$R = T - \mathbb{E}[T] - \big((S \odot M) - \mathbb{E}[S \odot M]\big)W_d,$$

so the stochastic gradient w.r.t. $W_d$ depends on the *centered* post-gating features $(S \odot M) - \mathbb{E}[S \odot M]$. By the matrix variance identity $\mathbb{E}[\varphi\varphi^\top] = \mathrm{Cov}(\varphi) + \mu\mu^\top$, removing the mean eliminates the rank-1 $\mu\mu^\top$ contribution that otherwise dominates the spectrum and couples to noise-induced shifts of $\mu$. Consequently, with PGB the effective gradient covariance is reduced from $\mathbb{E}[\varphi\varphi^\top]$ to $\mathrm{Cov}(\varphi)$, improving optimization stability.

*Remark* (Centering upstream reduces second moments). Suppose $\boldsymbol{X}$ has tokenwise expectations collected in rows of $\bar{\boldsymbol{X}}$ and let $\bar{\boldsymbol{\mu}}$ be the average row. Replacing $\boldsymbol{X}$ by $\boldsymbol{X} - \mathbf{1}\bar{\boldsymbol{\mu}}^{\top}$ strictly reduces $\mathbb{E}\|\boldsymbol{X}\|_F^2$ by $\|\bar{\boldsymbol{\mu}}\|_2^2$ per token. If $U = \boldsymbol{X}\boldsymbol{W}_u$ and $\boldsymbol{G} = \boldsymbol{X}\boldsymbol{W}_g$, this centering reduces $\mathbb{E}\|U\|_F^2$ and $\mathbb{E}\|\boldsymbol{G}\|_F^2$ by $\|\bar{\boldsymbol{\mu}}\boldsymbol{W}_u\|_2^2$ and $\|\bar{\boldsymbol{\mu}}\boldsymbol{W}_g\|_2^2$, respectively. Since SiLU is 1-Lipschitz and monotone, a first-order bound gives

$$\mathbb{E}\|\boldsymbol{S}\|_F^2 = \mathbb{E}\|\boldsymbol{U} \odot \mathrm{SiLU}(\boldsymbol{G})\|_F^2 \;\leq\; \mathbb{E}\|\boldsymbol{U}\|_F^2\, \mathbb{E}\|\,\mathrm{SiLU}(\boldsymbol{G})\|_\infty^2,$$

so upstream mean-removal lowers a surrogate bound on the second moment of $\boldsymbol{S}$. Trainable PGB lets the optimizer realize this reparametrization *in situ* by assigning baselines to $\boldsymbol{b}$ and reserving $\boldsymbol{U}, \boldsymbol{G}$ for deviations, thereby reducing the variance of $\boldsymbol{Y}$ and improving conditioning of $\boldsymbol{W}_d$.

**Stochastic perturbations from dropout.**   The analysis above shows that centering reduces the second moment of post-gating features in expectation. However, dropout (or other stochastic regularizers) reintroduces noisy offsets: even when $\mathbb{E}[\boldsymbol{S}] \approx 0$, the masked signal $\boldsymbol{M} \odot \boldsymbol{S}$ has expectation $p\boldsymbol{\mu}$ with variance inflated by $p(1-p)$. Thus, beyond deterministic centering, a trainable post-gating bias can serve as an *adaptive baseline* that cancels these noisy expectations. The following lemma formalizes this variance-reduction role.

### 3.3   VARIANCE REDUCTION UNDER DROPOUT

For sample $i$, the gradient contribution to $\boldsymbol{W}_d$ is

$$\frac{\partial \ell_i}{\partial \boldsymbol{W}_d} = \boldsymbol{y}_i \boldsymbol{\gamma}_i^{\top}, \qquad \boldsymbol{y}_i = \boldsymbol{M}_i \odot \boldsymbol{S}_i + \boldsymbol{b},$$

where $\boldsymbol{M}_i$ is a dropout mask, $\boldsymbol{S}_i = \boldsymbol{U}_i \odot \mathrm{SiLU}(\boldsymbol{G}_i)$, $\boldsymbol{\gamma}_i$ is the loss gradient, and $\boldsymbol{b}$ is the post-gating bias. Row-wise,

$$\left(\tfrac{\partial \ell_i}{\partial \boldsymbol{W}_d}\right)_{j:} = (\boldsymbol{M}_{ij}\boldsymbol{S}_{ij} + \boldsymbol{b}_j)\boldsymbol{\gamma}_i^{\top}.$$

Dropout intermittently zeros $\boldsymbol{S}_{ij}$, while a nonzero $\boldsymbol{b}_j$ supplies a stable baseline across all samples.

**Lemma 2** (Baseline minimizes gradient variance). *Assume $\mathbb{E}[\boldsymbol{S}_{ij}] = \boldsymbol{\mu}_j$, $\mathrm{Var}(\boldsymbol{S}_{ij}) = \sigma_j^2$, independence of $\boldsymbol{M}_{ij}$ and $\boldsymbol{\gamma}_i$, and $\mathbb{E}[\boldsymbol{\gamma}_i] = 0$. Then for any test direction $u$,*

$$\mathrm{Var}\!\left(u^{\top}\!\left(\tfrac{\partial \ell_i}{\partial \boldsymbol{W}_d}\right)_{j:}^{\top}\right) = \mathbb{E}[(\boldsymbol{u}^{\top}\boldsymbol{\gamma}_i)^2]\Big(p\sigma_j^2 + (p\boldsymbol{\mu}_j + \boldsymbol{b}_j)^2\Big),$$

*which is minimized uniquely at $\boldsymbol{b}_j^{\star} = -p\boldsymbol{\mu}_j$.*

*Interpretation.* The post-gating bias can cancel the masked mean $p\boldsymbol{\mu}_j$, leaving only the irreducible term $p\sigma_j^2$. Thus PGB centers each regressor and strictly reduces gradient variance. Without PGB, the mean must be encoded through masked features, inflating variance and worsening conditioning.

## 4   EXPERIMENTS AND RESULTS

To evaluate the effectiveness of post-gating bias (PGB), we tested the central hypothesis across three representative settings: (i) a Vision Transformer VAE on CelebA, where stochastic latent encodings create noisy training signals, (ii) small-scale language model pretraining on OpenWebText, where autoregressive objectives dominate, and (iii) stacked gated MLP classifiers on MNIST Xiao et al. (2017) and CIFAR-10*CIFAR-10/100* Krizhevsky (2009), where robustness to adversarial perturbations can be directly assessed. Together, these experiments probe whether the benefits of PGB are specific to noisy latent autoencoders or extend to other architectures.

**Experimental setting.**   For CelebA we use the standard training/validation/test splits. Our ViT-VAE employs $16 \times 16$ patches, an embedding dimension of 576, 12 attention heads, and 8 encoder and 8 decoder transformer blocks. Models are trained with Adam at a learning rate of $10^{-4}$ using a cosine schedule down to $10^{-5}$, with dropout fixed at 0.1. The loss combines RMSE reconstruction error with a latent length penalty, and Gaussian noise is added to latent codes during training.

For language modeling we pretrain Phi3-mini and Gemma3 from scratch on OpenWebText, again using standard splits. Each model is trained with Adam at $10^{-4}$ with the same cosine schedule,

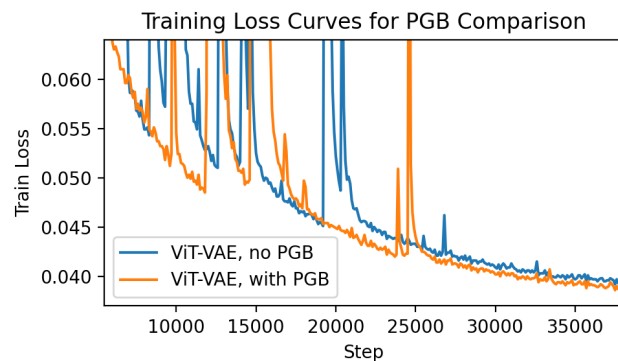

Figure 1: Training loss comparisons on a transformer-based VAE for CelebA. PGB shortens loss spikes and leads to longer stable regions of descent, with a slight asymptotic improvement.

and dropout probability 0.1. We evaluate test loss under four bias configurations: no bias, PGB only, RMSNorm bias only, and both biases. For Gemma3 we additionally modify the architecture to include dropout after the gated product, in order to test sensitivity to stochastic masking.

For gated MLPs we train 3-layer models with hidden width 128 on MNIST and CIFAR-10, using Adam at $10^{-4}$ and no dropout. In addition to clean test accuracy, we evaluate adversarial robustness using the fast gradient sign method (FGSM), applying uniform input noise of 0.2 and 10 attack steps with a step size 0.015.

### 4.1 ViT-VAE on CelebA

We first examine a Vision Transformer VAE on CelebA. Figure 1 shows training losses with and without a post-gating bias (PGB). At early epochs the two variants behave similarly. However, loss spikes occur sooner and persist longer in the model without PGB. By contrast, PGB shortens the duration of spikes, leading to longer regions of stable descent and ultimately a better asymptotic outcome.

Although the effect is modest, it is consistent with our analysis: PGB provides a direct intercept that can absorb noisy fluctuations in the latent signal, allowing gradients for $\boldsymbol{W}_d$ to be driven by centered features. Given that the implementation requires only an in-place addition and negligible parameter cost, the observed improvement highlights a potentially useful stabilization mechanism for transformer-based VAEs.

### 4.2 LLM Pretraining

We next consider large language model pretraining. Here we test Phi3-mini and Gemma3 with four bias configurations: (1) no biases, (2) PGB only, (3) RMSNorm bias only, and (4) both biases. The Gemma3 architecture is also modified to include dropout after the gated product.

Figures 2 and 3 show that none of these variants makes a significant difference in training loss. This supports the hypothesis that the main benefit of PGB is not in attention-heavy autoregressive settings, but rather in architectures where stochastic noise interacts with latent encodings, as in VAEs.

### 4.3 Stacked Gated MLPs on MNIST and CIFAR-10

Finally, we evaluated stacked Gated MLPs (without attention) on MNIST and CIFAR-10 under adversarial attack. Here an unexpected result emerged: adding bias after gating consistently improved robustness to fast gradient sign method (FGSM) perturbations.

Figure 4 shows adversarial test accuracy on CIFAR-10 with three hidden layers of width 128. The model with PGB maintains a clear advantage in adversarial accuracy throughout training. On MNIST, the effect is also visible. With three hidden layers of width 256 (Figure 5), PGB raises

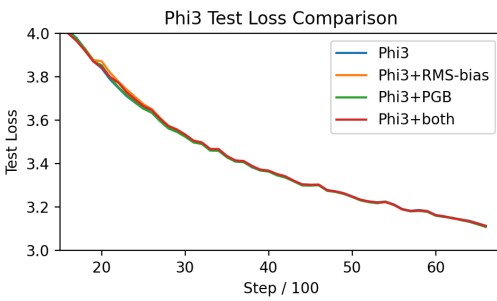 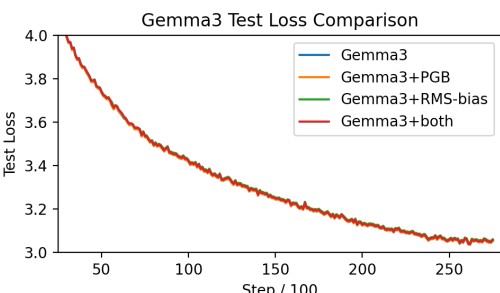

Figure 2: Test loss for Phi3 under four bias configurations. No significant effect is observed.

Figure 3: Test loss for Gemma3, including gated signal dropout, under four bias configurations. Again, no discernible effect.

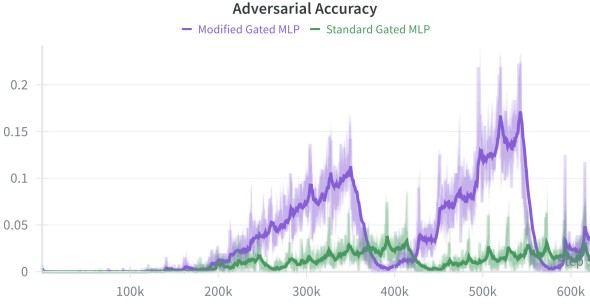

Figure 4: Adversarial accuracy during training of stacked Gated MLPs on CIFAR-10 (3 hidden layers, width 128). Models with PGB show stronger robustness to FGSM perturbations. All models are trained with Adam at learning rate $10^{-4}$ and no dropout.

adversarial accuracy from about 14% to 16%. Interestingly, in a smaller configuration with two hidden layers of width 64 (Figure 6), the benefit is much more pronounced, suggesting that PGB may provide stronger robustness in lower-capacity models.

Although preliminary, these results point to a potential role for explicit baselines in enhancing adversarial robustness in purely feed-forward gated networks. Further work is needed to test whether this effect generalizes beyond MNIST and CIFAR-10 and under stronger adversarial settings.

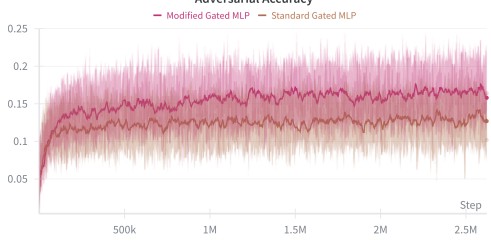 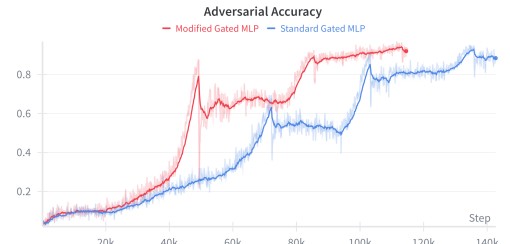

Figure 5: Adversarial accuracy on MNIST for stacked Gated MLPs with 3 hidden layers of width 256. Here PGB raises adversarial test accuracy from $\sim 14\%$ to $\sim 16\%$.

Figure 6: Adversarial accuracy on MNIST for stacked Gated MLPs with 2 hidden layers of width 64. PGB provides a pronounced robustness gain under FGSM perturbations compared to the bias-free baseline.

**Grokking Phenomenon.** Interestingly, the phenomenon of grokking, where neural networks gradually become more robust or generalize better long after achieving near-perfect training perfor-

mance, has been documented not only in algorithmic tasks but also in classification settings, including adversarial robustness (Humayun et al., 2024). In their formulation, delayed robustness arises as the network's partition of linear regions migrates in input space, reducing sensitivity around training points and pushing non-linearities toward decision boundaries. From this perspective, PGB might act as an accelerant or enabler of this transition: by stabilizing the baseline routing early, it may speed up or strengthen the migration dynamics that underlie robust grokking. In our experiments, we observe that adversarial accuracy with PGB begins rising earlier and stabilizes higher than the bias-free counterpart, consistent with the view that PGB helps the network "grok robustness" sooner or more reliably.

## 5 DISCUSSION

Our study began from an empirical observation: introducing a post-gating bias (PGB) in a transformer VAE yielded a marked improvement in training stability and asymptotic loss. Motivated by this finding, we analyzed the role of PGB as restoring affine freedom after multiplicative gating. The analysis shows that without an explicit intercept, weight-only recovery of constant offsets is ill-conditioned, especially when dropout or latent noise perturb the mean direction of post-gating features. A trainable PGB both improves conditioning (by removing a rank-1 term from the Hessian) and reduces variance in gradients (by centering the masked regressor under dropout).

Our experiments provide preliminary but informative evidence:

- In a ViT-VAE on CelebA, PGB shortens loss spikes and yields a slight asymptotic benefit, consistent with the variance-reduction interpretation.

- In two LLM architectures (Gemma3 and Phi3-mini), adding PGB or other biases had no discernible effect, suggesting that autoregressive pretraining is less sensitive to this mechanism.

- In stacked Gated MLPs, PGB improved robustness to adversarial perturbations, an effect that deserves further study.

Taken together, these findings suggest that PGB is not tied solely to gated dropout, as we initially suspected, but may be most relevant in settings where stochastic noise interacts with latent encodings. The VAE experiments point in this direction, as PGB provides a baseline that absorbs fluctuations otherwise routed through ill-conditioned weights. We also observed that training instabilities were most visible early in optimization, when the learning rate is large, but largely vanish once the rate decays. This interplay between gradient variance, noise level, and learning-rate dynamics deserves closer study.

The analysis highlights that the main role of PGB is geometric: by removing a rank-1 outer product from the effective Hessian, it reduces the dominance of a single high-variance direction. This is less about expressivity (which is unchanged) and more about conditioning. From this perspective, PGB is a form of variance control: it reshapes the eigen-spectrum so that small perturbations are not amplified along unstable directions, making optimization smoother.

More broadly, our results caution against the common view that biases in transformers are entirely redundant. While this heuristic holds in attention-heavy LLMs, our analysis and experiments reveal that redundancy breaks down after multiplicative gating. Thus, what appears redundant at the level of expressivity may not be redundant at the level of optimization dynamics. Architectural simplifications motivated solely by redundancy arguments should therefore be revisited when conditioning or robustness are at stake.

Here, even a negligible-cost bias can alter the conditioning of the optimization problem, stabilizing features that would otherwise be transmitted through high-variance channels. In this sense, PGB is less about expressivity and more about *geometry*: it reshapes the spectrum of the effective Hessian in a way that lowers gradient variance and smooths descent.

The connection to robustness is particularly intriguing. Adversarial experiments on MNIST and CIFAR-10 show that PGB increases resistance to perturbations, echoing recent arguments that robustness eventually emerges in overparameterized models through continued training. By supplying

a baseline earlier in optimization, PGB may accelerate this "grokking" of robustness or improve its asymptotic level, though confirming this will require systematic study.

Finally, we emphasize the practicality of the modification. In an era where the cost of architectural changes is heavily scrutinized, PGB is effectively free: it can be implemented as an in-place addition, introducing negligible parameter or memory overhead. This positions it as a candidate for inclusion wherever training instabilities or robustness concerns are present. Future work should test PGB more broadly: across multiple seeds and noise levels in VAEs, with spectral diagnostics that track the evolution of Hessian eigenvalues, and in domains where latent noise is inherent, such as diffusion autoencoders or scientific surrogate models.

In summary, PGB is not a universal fix, but it offers a small and elegant tool for mitigating the interaction between gating, stochastic noise, and optimization stability. Its simplicity makes it easy to adopt, and even modest improvements may prove valuable in practice.

# 6 CONCLUSION

We introduced Post-Gating Bias (PGB), a simple addition that restores affine freedom in transformer MLPs after multiplicative gating. Our analysis shows that PGB improves conditioning and reduces gradient variance under noise, without altering expressivity or incurring cost. Experiments confirm that its impact is modest in autoregressive pretraining but beneficial in VAEs and gated MLPs, where it stabilizes training and enhances robustness. These results highlight that even "redundant" components can matter for optimization dynamics, suggesting that bias placement in gated networks deserves renewed attention.

# 7 LIMITATIONS AND OUTLOOK

Our evaluation is limited in several respects. First, we have not presented the crystallographic VAE case that originally motivated this study, as it depends on an unpublished architecture. Second, our experimental trials are few in number, and further repetitions are needed to confirm robustness across seeds, hyperparameters, and training regimes. The absence of gains in LLM pretraining underscores the need to better delineate when and why PGB helps, and whether its role is confined to noisy latent encoders or extends more broadly.

A further limitation concerns adversarial robustness and grokking. While our stacked gated MLP experiments revealed improvements in adversarial accuracy, these results are preliminary and not yet fully understood. We observed interactions with architectural depth and width, suggesting that the benefits of PGB may depend on how baselines are represented across layers. Moreover, although the robustness gains are suggestive of earlier grokking, consistent with recent work showing that robustness eventually emerges with sufficient training, we cannot yet determine whether PGB accelerates this process, improves asymptotic robustness, or simply interacts with optimization noise in a coincidental way. Even in LLMs, the seemingly identical loss trajectories may conceal deeper differences in the learned representations that our current evaluation does not capture.

That said, we believe this work is a useful contribution: it highlights a simple modification, grounded in analysis, that can sometimes mitigate instability at effectively zero cost. We hope this encourages further experimentation in diverse architectures, particularly in noisy or generative settings where stable optimization remains a challenge.

**Use of LLMs.** We used ChatGPT (GPT-5, OpenAI) to assist with improving the clarity and consistency of exposition, formatting LaTeX, and checking grammar. All ideas, experiments, and analyses were conceived, designed, and executed by the authors.

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

# APPENDIX A    PROOFS FOR ANALYSIS SECTION

## A.1    PROOF OF LEMMA 1 (MINIMAL-CHANGE EXACT-FIT)

We solve
$$\min_{\boldsymbol{\Delta}} \|\boldsymbol{\Delta}\|_F \quad \text{s.t.} \quad \boldsymbol{\Delta} \boldsymbol{b}_{\text{in}} = \boldsymbol{b}_{\text{out}}.$$
Let $\boldsymbol{u} = \boldsymbol{b}_{\text{in}}/\|\boldsymbol{b}_{\text{in}}\|_2$. Any feasible $\boldsymbol{\Delta}$ must satisfy $\boldsymbol{\Delta} \boldsymbol{u} = \boldsymbol{b}_{\text{out}}/\|\boldsymbol{b}_{\text{in}}\|_2$. Among all linear maps that send $\boldsymbol{u}$ to this value, the one with smallest Frobenius norm is the rank-1 map
$$\boldsymbol{\Delta}^* = \frac{\boldsymbol{b}_{\text{out}} \, u^\top}{\|\boldsymbol{b}_{\text{in}}\|_2} = \frac{\boldsymbol{b}_{\text{out}} \, \boldsymbol{b}_{\text{in}}^\top}{\|\boldsymbol{b}_{\text{in}}\|_2^2}.$$
Uniqueness follows from orthogonality of the Frobenius norm: any additional component off $\text{span}(\boldsymbol{b}_{\text{in}})$ increases $\|\boldsymbol{\Delta}\|_F$ without improving feasibility. $\qquad\square$

## A.2    PROOF OF REMARK (MINIMUM-MSE PERTURBATION)

We minimize
$$\min_{\boldsymbol{\Delta}} \mathbb{E}\|(\boldsymbol{W}(\boldsymbol{x} + \boldsymbol{b}_{\text{in}}) + \boldsymbol{b}_{\text{out}}) - (\boldsymbol{W} + \boldsymbol{\Delta})(\boldsymbol{x} + \boldsymbol{b}_{\text{in}})\|_2^2,$$
where $\boldsymbol{x}$ has mean zero and covariance $\boldsymbol{\Sigma}$. Expanding,
$$\mathbb{E}\|\boldsymbol{W}(\boldsymbol{x} + \boldsymbol{b}_{\text{in}}) + \boldsymbol{b}_{\text{out}} - (\boldsymbol{W} + \boldsymbol{\Delta})(\boldsymbol{x} + \boldsymbol{b}_{\text{in}})\|_2^2 = \mathbb{E}\|(\boldsymbol{b}_{\text{out}} - \boldsymbol{\Delta}\boldsymbol{b}_{\text{in}}) - \boldsymbol{\Delta}\boldsymbol{x}\|_2^2.$$
Since $\mathbb{E}[\boldsymbol{x}] = 0$, this equals
$$\|\boldsymbol{b}_{\text{out}} - \boldsymbol{\Delta}\boldsymbol{b}_{\text{in}}\|_2^2 + \text{tr}(\boldsymbol{\Delta}\boldsymbol{\Sigma}\boldsymbol{\Delta}^\top).$$
This is a convex quadratic in $\boldsymbol{\Delta}$ with unique minimizer given by normal equations
$$(\boldsymbol{\Sigma} + \boldsymbol{b}_{\text{in}}\boldsymbol{b}_{\text{in}}^\top)\boldsymbol{\Delta}^\top = \boldsymbol{b}_{\text{in}}\boldsymbol{b}_{\text{out}}^\top,$$
hence
$$\boldsymbol{\Delta}^* = \boldsymbol{b}_{\text{out}} \, \boldsymbol{b}_{\text{in}}^\top(\boldsymbol{\Sigma} + \boldsymbol{b}_{\text{in}}\boldsymbol{b}_{\text{in}}^\top)^{-1}.$$
Substituting back yields the minimal MSE formula in the main text. $\qquad\square$

## A.3    PROOF OF PROPOSITION 1 (INTERCEPT REMOVAL IMPROVES CONDITIONING)

Let $\boldsymbol{\varphi} \in \mathbb{R}^p$ denote post-gating features with mean $\boldsymbol{\mu}$ and covariance $\boldsymbol{\Sigma}$. The uncentered Hessian is
$$\boldsymbol{H}_{\text{no-bias}} = \mathbb{E}[\boldsymbol{\varphi}\boldsymbol{\varphi}^\top] = \boldsymbol{\Sigma} + \boldsymbol{\mu}\boldsymbol{\mu}^\top,$$
while with explicit bias the Hessian reduces to $\boldsymbol{H}_{\text{pgb}} = \boldsymbol{\Sigma}$.

By the Rayleigh quotient, for the normalized top eigenvector $\mathbf{v}$ of $\boldsymbol{\Sigma}$,
$$\lambda_{\max}(\boldsymbol{H}_{\text{no-bias}}) \geq \mathbf{v}^\top(\boldsymbol{\Sigma} + \boldsymbol{\mu}\boldsymbol{\mu}^\top)\mathbf{v} = \lambda_{\max}(\boldsymbol{\Sigma}) + (\mathbf{v}^\top\boldsymbol{\mu})^2.$$
Thus $\lambda_{\max}(\boldsymbol{H}_{\text{no-bias}}) \geq \lambda_{\max}(\boldsymbol{\Sigma})$, with strict inequality when $\boldsymbol{\mu} \notin \ker(\boldsymbol{\Sigma})$. Since both Hessians share the same nullspace, the condition number satisfies
$$\kappa(\boldsymbol{H}_{\text{no-bias}}) \geq \kappa(\boldsymbol{\Sigma}),$$
with strict inequality unless $\boldsymbol{\mu} \in \ker(\boldsymbol{\Sigma})$. $\qquad\square$

## A.4    PROOF OF REMARK (CENTERING UPSTREAM REDUCES SECOND MOMENTS)

Suppose $\boldsymbol{X} \in \mathbb{R}^{n \times d}$ has tokenwise expectations $\bar{\boldsymbol{X}}$ (each row mean of $\boldsymbol{X}$), with global mean $\bar{\boldsymbol{\mu}} = \frac{1}{n}\mathbf{1}^\top\bar{\boldsymbol{X}}$. Replacing $\boldsymbol{X}$ by $\boldsymbol{X}' = \boldsymbol{X} - \mathbf{1}\bar{\boldsymbol{\mu}}^\top$ changes the Frobenius norm:
$$\mathbb{E}\|\boldsymbol{X}'\|_F^2 = \mathbb{E}\|\boldsymbol{X}\|_F^2 - n\|\bar{\boldsymbol{\mu}}\|_2^2.$$
Thus the expected squared norm is strictly reduced by $\|\bar{\boldsymbol{\mu}}\|_2^2$ per token.

If $\boldsymbol{U} = \boldsymbol{X}\boldsymbol{W}_u$ and $\boldsymbol{G} = \boldsymbol{X}\boldsymbol{W}_g$, then
$$\mathbb{E}\|\boldsymbol{U}'\|_F^2 = \mathbb{E}\|\boldsymbol{U}\|_F^2 - \|\bar{\boldsymbol{\mu}}\boldsymbol{W}_u\|_2^2, \qquad \mathbb{E}\|\boldsymbol{G}'\|_F^2 = \mathbb{E}\|\boldsymbol{G}\|_F^2 - \|\bar{\boldsymbol{\mu}}\boldsymbol{W}_g\|_2^2.$$
Since SiLU is 1-Lipschitz and monotone,
$$\mathbb{E}\|\boldsymbol{S}'\|_F^2 = \mathbb{E}\|\boldsymbol{U}' \odot \text{SiLU}(\boldsymbol{G}')\|_F^2 \leq \mathbb{E}\|\boldsymbol{U}'\|_F^2 \, \mathbb{E}\|\text{SiLU}(\boldsymbol{G}')\|_\infty^2,$$
which bounds the second moment of $S$ after centering. $\qquad\square$

## A.5   PROOF OF LEMMA 2 (BASELINE MINIMIZES GRADIENT VARIANCE)

For row $j$, define

$$\boldsymbol{g}_i(j, u) = \boldsymbol{u}^\top \left( \frac{\partial \ell_i}{\partial \boldsymbol{W}_d} \right)^\top_{j:} = (\boldsymbol{M}_{ij} \boldsymbol{S}_{ij} + \boldsymbol{b}_j)\, r_i, \qquad r_i = \boldsymbol{u}^\top \boldsymbol{\gamma}_i.$$

By independence and $\mathbb{E}[\boldsymbol{\gamma}_i] = 0$,

$$\mathrm{Var}(\boldsymbol{g}_i(j, u)) = \mathbb{E}[r_i^2]\, \mathrm{Var}(\boldsymbol{M}_{ij} \boldsymbol{S}_{ij} + \boldsymbol{b}_j).$$

Now,

$$\mathbb{E}[\boldsymbol{M}_{ij} \boldsymbol{S}_{ij}] = p\boldsymbol{\mu}_j, \qquad \mathrm{Var}(\boldsymbol{M}_{ij} \boldsymbol{S}_{ij}) = p\sigma_j^2 + p(1-p)\boldsymbol{\mu}_j^2.$$

Hence

$$\mathrm{Var}(\boldsymbol{M}_{ij} \boldsymbol{S}_{ij} + \boldsymbol{b}_j) = p\sigma_j^2 + (p\boldsymbol{\mu}_j + \boldsymbol{b}_j)^2,$$

and therefore

$$\mathrm{Var}(g_i(j, u)) = \mathbb{E}[r_i^2]\big(p\sigma_j^2 + (p\boldsymbol{\mu}_j + \boldsymbol{b}_j)^2\big).$$

This quadratic in $\boldsymbol{b}_j$ is uniquely minimized at $\boldsymbol{b}_j^\star = -p\boldsymbol{\mu}_j$, with minimum value $\mathbb{E}[r_i^2]\, p\sigma_j^2$.   $\square$

