# OpenReview forum: "Post-Gating Bias: Restoring Affine Freedom in Transformer MLPs"
_ICLR.cc/2026/Conference — Submitted to ICLR 2026_

### Official Review · Reviewer_48XY · 2025-10-28

**Soundness:** 2
**Presentation:** 3
**Contribution:** 2
**Rating:** 4
**Confidence:** 3

**Summary:**

This paper proposes Post-Gating Bias, which is an additive bias inserted after the SwiGLU product and before the down-projection, to “restore affine freedom” lost by multiplicative gating, arguing this improves conditioning and reduces gradient variance in noisy regimes.

**Strengths:**

- The intervention is precisely defined by equations (1), (2), (3), (4), can be done in-place, and adds virtually no parameters or memory. This makes adoption easy and the ablation surface clean.

- The paper explains why output-bias reconstructability fails after multiplicative gating and shows that PGB effectively removes a rank-1 outer-product from the Hessian, tightening the conditioning bound.

- Results deliberately span regimes with/without stochastic latent noise: ViT-VAE shows shorter loss spikes and slightly better asymptotics; LLM pretraining shows no effect; gated-MLPs gain adversarial robustness.

**Weaknesses:**

Overall, even with the caveats in the limitations, the contribution feels pretty narrow and incremental.

- Section 2 is short and a bit selective—it covers GLUs and VAEs but doesn’t really place PGB in the bigger picture (bias placement in Transformers, centering/variance reduction, normalization choices). A deeper survey with more citations would help the case for novelty.

- The paper admits there aren’t many runs, and there’s no gain in LLM pretraining. It’s still unclear when PGB reliably helps or how it scales. More seeds, bigger models/data, more tasks, and compute-matched baselines would make this stronger.

- “FGSM with 10 steps (step size 0.015) + uniform input noise 0.2” sounds more like multi-step PGD than one-step FGSM. Please spell out the threat model (norm/$\varepsilon$), step schedule, restarts, and the exact eval protocol so results are comparable.

- No supplementary material was submitted; the paper includes a brief 1–2-page appendix with the necessary theoretical verifications, but no code is provided.

**Questions:**

- Please consider expanding Section 2 to position PGB against prior work on bias placement in Transformers, centering/variance-reduction, and normalization choices, and explain what’s truly new relative to those lines?

- Please add compute-matched, multi-seed results across a couple of larger models/datasets and report a scaling trend so we know when PGB reliably helps (and when it doesn’t)?

- For robustness, was the attack FGSM or multi-step PGD? Please specify the threat model (norm/$\varepsilon$), step size, number of steps, restarts, and dataset splits, and include a standard baseline (e.g., PGD-20 or AutoAttack) for comparability.

---

### Official Review · Reviewer_H1At · 2025-10-28

**Soundness:** 1
**Presentation:** 1
**Contribution:** 1
**Rating:** 2
**Confidence:** 4

**Summary:**

The paper studies the problem of how gating can suppress bias term information. It presents some theoretical analysis demonstrating this and how this may lead to unstable training. It then investigates how this issue can be rectified by a simple low-cost fix of adding a bias parameter after gating (post-gating bias or PGB). It presents some experiments evaluating its effect on training stability and adversarial robustness, showing that it can be useful for ViT-VAE models while the effects on language models are negligible.

**Strengths:**

The paper presents some theoretical analysis showing how gating can suppress bias terms that may lead to unstable training, and then presents a simple low-cost fix of adding a bias parameter after gating.

**Weaknesses:**

The are several major weaknesses.

### 1. The main experiments are not convincing enough to support the claims that the proposed fix PGB stabilizes training and improves adversarial robustness.

First, the experiments in the paper are very limited. There is no gap for language models (Phi3, Gemma3). There is only one experiment showing results for ViT-VAE. Based on this, the authors conclude that PGB is beneficial “in architectures where stochastic noice interacts with latent encodings, as in VAEs”. This claim needs to be better supported with more experiments and ablations. The experiments on adversarial robustness are done using MLPs on CIFAR-10 or MNIST datasets, which may be sufficient to support theory, but are very simplistic to argue about performance gains.

Second, the results in Fig. 1 comparing ViT-VAE train loss with and without PGB look very similar. The differences are not significant enough to support the effectiveness of PGB.

Third, the adversarial robustness experiments show some gains with PGB, however, the accuracies are still quite poor.

In summary, the effectiveness of PGB is not well-supported.

### 2. The paper contributions are not significant enough, and it is not written well overall.

First, the considered problem is very niche and not very well-motivated. Section 2 states observations from prior work and motivates this work stating “none have revisited the role of bias placement relative to dropout and gating”, which can justify the novelty but is not a strong motivation for the work itself.

Further, the theoretical analysis itself seems quite simplistic. For instance, Lemma 1 is a very standard result and doesn’t need to be stated as a Lemma. It should be cited from relevant prior work.

These factors, coupled with the unconvincing experiments, makes the significance of the contributions questionable.

Second, several parts of the paper read like it’s an in-progress work and not a finished one. For instance, the use of “working hypothesis” in the abstract, discussion of some experimental results in the section on Background and Related Work, omitted hyperlinks, e.g., in line 107, missing definitions of some terms in line 140, the phrase “an unexpected result emerged” in line 320, etc. Further, the figures are not made properly, especially Figs. 4-6 which are missing axis labels. Additionally, there is a lot of repetition in Sections 5 and 7, both of seem to be added to cover the additional pages due to lack of more results and contributions, rather than adding any important information.

**Questions:**

Please see the weaknesses section above.

---

### Official Review · Reviewer_29dQ · 2025-10-31

**Soundness:** 3
**Presentation:** 3
**Contribution:** 2
**Rating:** 4
**Confidence:** 3

**Summary:**

This paper revisits the common practice of omitting additive biases in transformer MLPs, focusing on the SwiGLU architecture where multiplicative gating destroys constant offsets and thus removes affine freedom after the nonlinearity. The authors propose a simple modification: Post-Gating Bias (PGB), which adds a bias term after the gated product and before the down-projection. Through theoretical analysis and empirical studies across Vision Transformer VAEs, language models, and gated MLP classifiers, the paper demonstrates that PGB can improve training stability and adversarial robustness in noisy or regularized settings, while having negligible effect in standard large-scale language model pretraining. The work highlights subtle but important roles for architectural elements often considered redundant.

**Strengths:**

(1) The paper provides a solid theoretical justification for why biases are not always redundant in transformer blocks, especially after multiplicative gating, and analyzes the conditioning and variance reduction benefits of PGB.
(2) The proposed PGB modification is easy to implement, incurs negligible computational or memory cost, and can be adopted in existing architectures with minimal efforts.
(3) The authors evaluate PGB across diverse settings (ViT-VAEs, LLM pretraining, and stacked gated MLPs), providing evidence for its benefits and limitations.

**Weaknesses:**

(1) From the experiments, we can see that the improvements from PGB are only visible in specific models (e.g., ViT-VAEs, stacked gated MLPs), and there is no improvement in standard LLM pretraining (see Figure 2 and 3 from the paper). This would limit the significance of the proposed method, because I feel that LLM is generally more widely used for transformer models than the other two models. Another model type that is interesting is vanilla ViT (see [1] from the list below). This paper would be strengthened if some experiments on vanilla ViT is included, because it's a crucial componet in many vision-language models.

(2) The experiments on ViT-VAE showed some improvements, but it looks like the difference is quite small in terms of training loss. The authors mentioned that they used standard training/valiadtion/test splits for the CelebA dataset. However, only training loss is showed in the paper (see Figure 1). In addition, the paper does not investigate whether PGB leads to qualitatively different learned representations or downstream behaviors.

Overall, I agree that the proposed method is interesting and worth investigating. But the experiments could be strengthened to better demonstrate the real benefits of the proposed method in practice, e.g., LLM, vanilla ViT which are popular settings where transformers are used.

**Reference**:
[1] Dosovitskiy, Alexey. "An image is worth 16x16 words: Transformers for image recognition at scale." arXiv preprint arXiv:2010.11929 (2020).

**Questions:**

see my comments above

---

### Official Review · Reviewer_38f8 · 2025-11-01

**Soundness:** 1
**Presentation:** 2
**Contribution:** 1
**Rating:** 2
**Confidence:** 4

**Summary:**

This paper proposes use of a bias term after multiplicative gating in gated neural networks, with the argument that this bias is hard to be absorbed by other parameters and normalization layers.  Variance and stability arguments are presented to suggest that, while the weight matrix that comes after gating can in theory learn to account for this bias, it is not a good idea to rely on it to do so.  Gated networks with such bias are evaluated in three scenarios: a VAE setup using vision transformer, stacked gated MLPs, and autoregressive pre-training.

**Strengths:**

* Theoretical and empirical study of post-gating bias in gated neural networks

**Weaknesses:**

The empirical support for the proposed bias is quite weak for a few reasons:
* In the ViT-VAE scenario the only evidence for value of bias is in Figure 1 which shows training loss over time.  Subjective observations are made on onset and duration of spikes in training loss, but looking at the picture one could argue the other way around as well.  A more quantitative analysis and comparison needs to be carried out.
* The bias shows no effect in the pre-training scenario.
* In the stacked gated MLP scenario, only adversarial robustness is discussed.  However, what is the model performance as compared to baseline models (including state of the art pre-trained/fine-tuned models) at the baseline classification tasks? The observed adversarial robustness is interesting but both baseline performance and adversarial robustness needs to be contrasted with state of the art pre-trained/fine-tuned models.

**Questions:**

n/a

---

### Meta-Review · Area_Chair_NppH · 2026-01-06

**Summary:**

This paper proposes Post-Gating Bias (PGB), an additive bias inserted after the SwiGLU (multiplicative) product and before the down-projection in gated networks. The motivation is that multiplicative gating can suppress affine freedom and interfere with output-bias reconstructability, potentially harming conditioning and gradient variance, especially in noisy or regularized regimes. The authors present a theoretical analysis showing that PGB can reduce rank-1 components in the Hessian and improve stability, and they provide empirical results on ViT-VAEs, stacked gated MLPs, and LLM pretraining.

**Reviewer Concerns:**

Three negative reviews.

**Reviewer Scores:**

Three negative reviews. There is no rebuttal.

---

### Decision · Program_Chairs · 2026-01-26

Reject